# Structural order in plasmonic superlattices

Florian Schulz [1,2✉], Ondřej Pavelka [3], Felix Lehmkühler [2,4], Fabian Westermeier [4], Yu Okamura [5], Niclas S. Mueller [5], Stephanie Reich [5] & Holger Lange [1,2]

The assembly of plasmonic nanoparticles into ordered 2D- and 3D-superlattices could pave the way towards new tailored materials for plasmonic sensing, photocatalysis and manipulation of light on the nanoscale. The properties of such materials strongly depend on their geometry, and accordingly straightforward protocols to obtain precise plasmonic super-lattices are highly desirable. Here, we synthesize large areas of crystalline mono-, bi- and multilayers of gold nanoparticles >20 nm with a small number of defects. The superlattices can be described as hexagonal crystals with standard deviations of the lattice parameter below 1%. The periodic arrangement within the superlattices leads to new well-defined collective plasmon-polariton modes. The general level of achieved superlattice quality will be of benefit for a broad range of applications, ranging from fundamental studies of light–matter interaction to optical metamaterials and substrates for surface-enhanced spectroscopies.

[1] Institute of Physical Chemistry, University of Hamburg, Grindelallee 117, 20146 Hamburg, Germany. [2] The Hamburg Centre for Ultrafast Imaging (CUI), Luruper Chaussee 149, 22761 Hamburg, Germany. [3] Department of Chemical Physics and Optics, Charles University, Ke Karlovu 3, 121 16 Prague 2, Czech Republic. [4] Deutsches Elektronen-Synchrotron DESY, Notkestr. 85, 22607 Hamburg, Germany. [5] Department of Physics, Freie Universität Berlin, Arnimallee 14, D-14195 Berlin, Germany. ✉email: Florian.Schulz@chemie.uni-hamburg.de

Ordered arrays and superlattices of nanoparticles (NPs) have a huge potential for new materials with tuneable electrical, optical, and mechanical properties[1–13]. Common strategies to produce such structures include lithography-based nanofabrication[14–16], and self-assembly of NPs functionalized with polymers[10], DNA[17–22], or dendritic molecules[23,24]. The ability to control both, the structure of the NP building blocks and the geometry of the superstructure might lead to designer materials with, for example, topological states and Dirac minibands in the case of semiconductor NPs[25,26]. However, the transition from localized to the required delocalized electrons is often hindered by the lack of periodic order, size variations and, most importantly, the insufficient degree of electronic coupling between the NPs[27]. This can in part be circumvented by coupling the electromagnetic (dipole) fields instead of electrons. Then, scale and order of the superlattice can lead to new collective properties[28–30]. For example, the superfluorescence from periodically coupled excitonic dipoles of perovskite NPs was recently demonstrated[31]. In similar ways, plasmonic dipoles of metal NPs interact via dipole–dipole interaction[30,32–34]. This creates a continuum of collective plasmonic states where the electrons of the metal NP crystal remain localized. According to recent experiments, the coupling strength of such periodically coupled plasmons to photons can become comparable to the photon energy[35]. The optical properties are expected to be widely tuneable through crystal symmetry and geometry[35–37].

A prerequisite for the formation of well-defined coupled modes is the uniformity of the constituents and a high degree of order on relevant scales (optical wavelength). The coupling strength increases with decreasing interparticle spacings, gaps <10 nm are favorable[5,35,38]. Upon today, it has been difficult to combine a high particle uniformity (size and shape) with a high degree of periodic order and large crystalline areas in the desired geometry. For small NPs (diameters < 15 nm), a huge variety of mesoscale superlattices have been reported[6,10–12,39–51]. However, most reports of these NP superlattices focus on the structural and electronic properties. The few optical studies exploiting the periodic order of plasmonic components, e.g. use lithographic structures as waveguides, where plasmons couple over much longer distances[34,52]. The reason for the lack of reports on periodic plasmon coupling is that there exists no robust protocol to assemble large plasmonic NPs like gold NPs (AuNPs) into large-scale crystalline superlattices. The plasmonic dipole strength, which is relevant for the periodic coupling, scales with particle size, yielding small AuNPs less attractive for periodic dipole field coupling. For example, to observe well-defined plasmon polaritons, the minimum AuNP diameter is ~25 nm[36,53,54].

In this work, we present a robust protocol for assembling large (diameter > 20 nm) AuNPs into crystalline superlattices up to >0.01 mm² for monolayers and even larger crystalline bilayer and multilayer with interparticle gaps in the range of 1–8 nm and standard deviations of the lattice constant of below 1%. This degree of order allows well-defined collective plasmon-polariton modes to emerge: we compare the film properties with the ones of less ordered, conventional films of comparable scales and demonstrate how order leads to the formation of new polaritonic excitations.

## Results

**AuNP synthesis and functionalization**. Periodically coupled plasmonic modes require a continuous lattice of uniform NPs. NP size and shape deviations are disadvantageous for two reasons: they result in local variations of the (size-dependent) plasmonic properties and reduce the periodicity of the NP superlattice. To reproducibly obtain superlattices with interparticle gaps below 10 nm,

our strategy therefore was to synthesize AuNPs with optimized dispersity and uniformity using the cetyltrimethylammonium chloride (CTAC)-based seeded growth strategy presented by Zheng et al.[55]. The AuNPs were functionalized with polystyrene-thiol ligands (PSSH) that have sufficiently low molecular weights (~2000, 5000, and 12,000 g mol⁻¹; PSSH2k, PSSH5k, and PSSH12k) to allow for small interparticle gaps. At the same time, they still provide the stabilization that is necessary for high particle concentrations and stable assemblies in the desired diameter range. In a previous study we tested the potential of PSSH-promoted self-assembly with citrate-stabilized AuNPs (AuNP@Citrate) with dispersities (rsd) of ~8%. Despite some promising results, the limitations discussed above were encountered[56]. AuNP@CTAC have a more challenging surface chemistry and are obtained in lower yields than AuNP@Citrate but exhibit superior uniformity and lower dispersity. The CTAC-based protocol was scaled up by a factor of 10 to obtain sufficient final concentrations. Phase-transfer-based ligand exchange as described for AuNP@Citrate does not work for AuNP@CTAC[56]. Therefore, AuNP@CTAC were functionalized with PSSH by a ligand exchange in THF, transferred to toluene, purified and concentrated for self-assembly experiments. The ligand exchange and the colloidal stability of the functionalized AuNPs were monitored by UV/Vis spectroscopy.

AuNP superlattice films were prepared by evaporation of AuNP solutions in toluene on a liquid subphase: diethylene glycol (DEG), as presented by Dong et al. for smaller NPs[57]. A scheme and resulting exemplary superlattice films are shown in Fig. 1. The straightforward protocol starts by pipetting of the AuNPs in toluene onto the DEG subphase in a teflon or high density polyethylene (HDPE) well (Fig. 1a). The well is then covered with a cover slip and left undisturbed. Superlattice films usually form within 16–24 h (Fig. 1b) and can then be transferred onto arbitrary substrates like transmission electron microscopy (TEM) grids, glass slides, or silicon wafers. The superlattices can also form freestanding films (~40 μm × 40 μm, Supplementary Fig. 1).

All AuNP samples formed well-ordered AuNP superlattices, differing in the number of layers ranging from monolayer to multilayer (>20), number of defects, crystallite size and the crystal mosaicity (Fig. 1c–g and Supplementary Figs. 2–11). For all superlattice films, defect-free monolayers in the μm² range were routinely observed (Fig. 1 and Supplementary Figs. 2–11) up to monolayers with long-range periodic order over areas >10,000 μm². Even larger areas were obtained for crystalline bilayers (Fig. 1f and Supplementary Figs. 2–4, 6, 10, 11). Table 1 lists the studied samples with the concentrations $c$ used for superlattice preparation and the interparticle edge-to-edge distances (gaps) in the formed superlattices determined by TEM and small-angle x-ray scattering (SAXS).

The gaps obtained by SAXS deviate up to a few nm from those obtained by TEM analysis. This observation points at the limits of the TEM analysis accuracy, especially for the larger AuNP which are slightly less uniform (cf. Supplementary Figs. 2–11)[58]. Consequently, complementary SAXS measurements for comparison are recommendable for the characterization of superlattices in general. AuNP superlattices covering large areas have been reported in several pioneering studies and recent reviews are available[10,39–47,59,60], but the crystalline order presented herein is unprecedented to the best of our knowledge. With increasing number of layers, the superlattices can be described as well-defined AuNP-supercrystals (Fig. 1g and Supplementary Fig. 12). The number of layers cannot be controlled perfectly, but as a tendency slow evaporation and an increasing number of particles in the drying solution seem to favor higher fractions of multilayer superlattices. Accordingly, a fast evaporation and low AuNP concentrations favor the formation of monolayers. Quantitative statements are not possible because the exact fractions of the

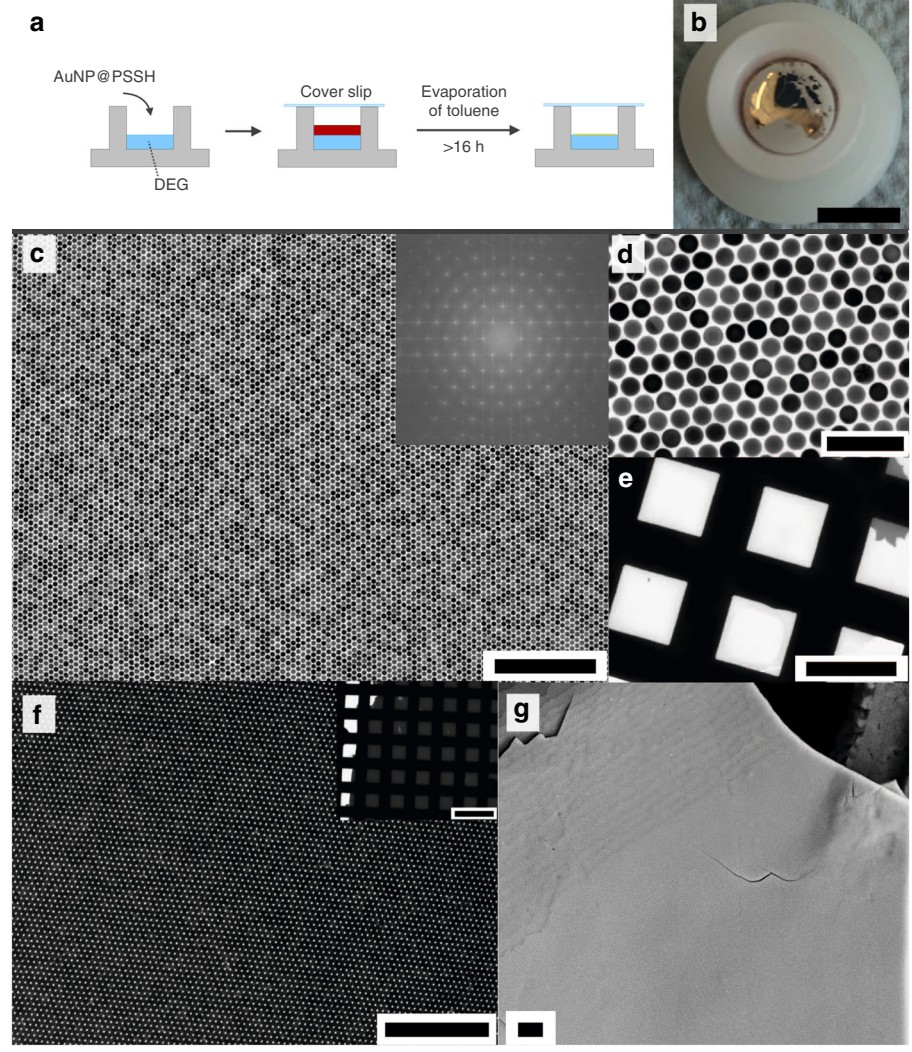

**Fig. 1 AuNP superlattice synthesis. a** Scheme of superlattice preparation. First, 100 µl of AuNP@PSSH (sample details in Table 1) in toluene are pipetted onto a liquid subphase (DEG), then the well is covered with a glass slip to slow down the evaporation of the toluene. Superlattice films typically form within 16–24 h. **b** Photograph of a formed superlattice, scalebar 1 cm. **c**, **d** TEM measurements at different magnifications of a monolayer area in superlattices formed from AuNP25@PSSH2k (**c**: scalebar 500 nm, **d**: scalebar 100 nm). The inset in **c** shows the FFT of the digital image. **e** Low magnification of the same crystalline monolayer area covering >0.01 mm² of the TEM grid (scalebar: 50 µm). The small darker gray area in the top right corner is a bilayer. **f** Bilayer with few defects formed from AuNP25@PSSH5k (scalebar 500 nm). The inset shows an extended bilayer area (scalebar 100 µm). **g** SEM measurements of a crystalline multilayer superlattice with 22–23 layers formed from AuNP35@PSSH12k (scalebar 2 µm).

**Table 1 AuNP samples used for film formation and resulting interparticle distances (gap).**

| Sample name | AuNP diameter (nm)[a] | Ligand | c(AuNP) (nM) | Gap (TEM) (nm)[b] | Gap (SAXS)[c] |
|---|---|---|---|---|---|
| AuNP25@PSSH2k | (25.1 ± 0.7) | PSSH2k | 6.3 | 3.0 | 3.8 nm |
| AuNP25@PSSH5k | (25.1 ± 0.7) | PSSH5k | 29.6 | 3.3 | 4.3 nm |
| AuNP25@PSSH12k | (25.1 ± 0.7) | PSSH12k | 30.0 | 7.7 | NA |
| AuNP35@PSSH5k | (37.3 ± 1.2) | PSSH5k | 1.6 | 5.1 | NA |
| AuNP35@PSSH12k | (37.3 ± 1.2) | PSSH12k | 1.6 | 5.2 | 6.3 nm |
| AuNP50@PSSH5k | (50.5 ± 1.1) | PSSH5k | 0.7 | 1.3 | 3.6 nm |
| AuNP50@PSSH12k | (50.5 ± 1.1) | PSSH12k | 0.7 | 2.7 | NA |
| AuNP60@PSSH2k | (63.4 ± 1.8) | PSSH2k | 0.3 | 1.2 | NA |
| AuNP60@PSSH5k | (63.4 ± 1.8) | PSSH5k | 0.3 | 2.2 | NA |
| AuNP60@PSSH12k | (63.4 ± 1.8) | PSSH12k | 0.3 | 1.3 | 4.3 nm |

[a]Core diameter by TEM, with standard deviation of the mean ($N > 200$).
[b]By nearest neighbor analysis of TEM measurements.
[c]Obtained by analysis of SAXS measurements as discussed in the text.

different layer numbers within one sample are not accessible. A detailed description of the synthesis and relevant parameters is provided in the "Methods" section.

An important advantage of using grafted polymers for film formation is that they allow controlling the interparticle distance by changing their molecular weight[10–12,39,43,45,56,61]. However, although the PSSH ligand length was found to affect the interparticle distances (gaps), a clear tendency was not always observed and the correlation was not linear. For PSSH12k a correlation of gap distance with particle diameter was observed that points to the increasing attractive van-der-Waals-potentials between the AuNP-cores (Supplementary Fig. 13)[62]. However, with PSSH5k the trend was not that clear, indicating that additional parameters of the crystallization affect the gaps. For PSSH2k just two samples can be compared due to the limited stabilization provided by this small ligand. In the range of particle diameters and ligands tested, the gap spacings varied from 1 to 8 nm (Table 1, Supplementary Fig. 13), an ideal range for periodic plasmonic coupling that has been challenging to achieve with high precision[5].

For a quantitative measure of the superlattice crystallinity, scanning microfocus SAXS measurements (beamsize 2.8 μm × 1.7 μm ($h \times v$)) were performed on selected AuNP superlattice films (Fig. 2) in transmission mode. We measured SAXS patterns on hundreds of different spots on each sample including thin (monolayer and bilayer) and thick regions (multilayers). An exemplary intensity map is shown in Fig. 2a, where each pixel represents the measured intensity in a certain $q$-range of a single SAXS measurement. The 2D-scattering pattern and azimuthally averaged $I(q)$ (Fig. 2b) exhibit pronounced Bragg peaks of a six-fold order which can be well described with scattering from a 2D-hexagonal lattice. This confirms the periodic order and allows extracting the lattice constant $a$. The distributions $P(a)$ of the lattice constant $a$ for different samples are shown in Fig. 2c. The exceptionally narrow distributions of $a$ underline the robust and long-range periodic order within the superlattices. In comparison, the distribution for an AuNP@Citrate-based superlattice (30 nm, functionalized with PSSH5k) is 1.5–6 times broader (Fig. 2d, gray dashed line). Remarkably, the small lattice constant difference (~0.5 nm) of AuNP25@PSSH2k and −5k due to different ligand lengths can be clearly discerned (Fig. 2c) and agrees well with the analysis of TEM measurements (Table 1).

The achieved large superlattice areas in combination with the possible deposition on transparent substrates enable the investigation of the plasmonic properties by optical microscopy and spectroscopy. The number of layers can be clearly distinguished in optical transmittance (Fig. 3a, Supplementary Fig. 14) and the layer-number-dependent optical properties can be measured with spectroscopy.

Figure 3b shows the optical absorbance of a monolayer, bilayer, and trilayer of periodically arranged AuNPs. Pronounced absorbance peaks appear in the NIR spectral range from the direct optical excitation of plasmon polaritons that form standing waves in the supercrystals (see insets)[35,36,63]. With each additional layer a new polaritonic mode is activated, starting from a single mode for a bilayer[35,36]. The excitation of plasmon-polaritons leads to pronounced reflectance dips and transmittance maxima in the optical spectra (Fig. 3c). A polaritonic stop band prevents light from entering the supercrystal in the visible spectral range and light is mostly reflected[35]. The reflectance decreases for excitation energies larger than 2 eV because of the interband transitions of gold. The optical spectra are in good agreement with finite-difference time-domain (FDTD) simulations (compare solid and dashed lines in Fig. 3c), demonstrating the control of the structural parameters in the synthesis process. The number, energy, and spectral width of the polaritonic modes depends sensitively on the interparticle

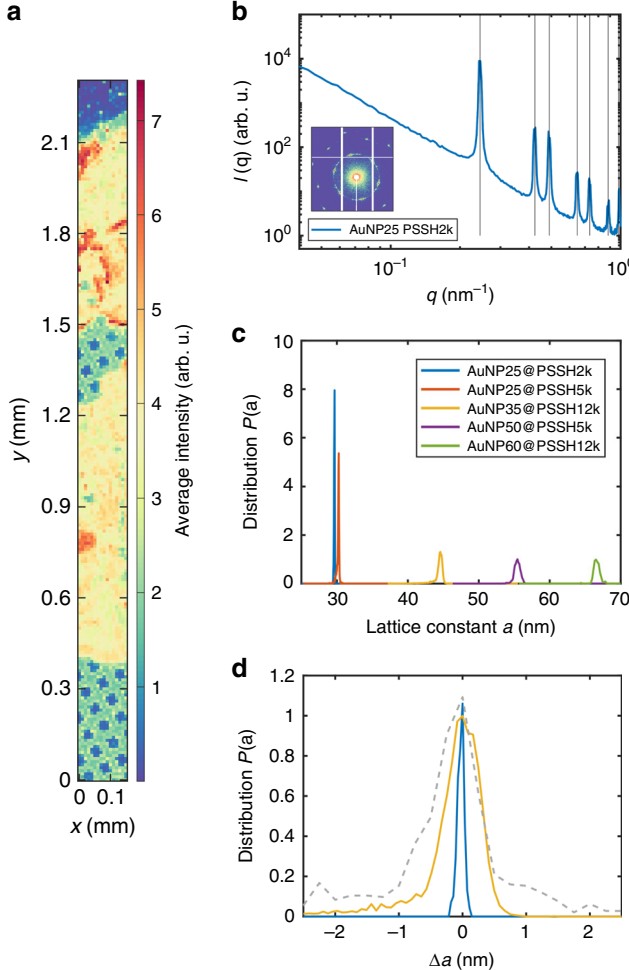

**Fig. 2 SAXS investigation of AuNP superlattices. a** Exemplary intensity map (AuNP25@PSSH2k) obtained by scanning SAXS with 10 μm step width where each pixel corresponds to one SAXS measurement. The structure of the copper grid can be discerned. **b** Exemplary $I(q)$ of a single SAXS pattern and 2D scattering pattern (logarithmic intensity scale) for AuNP25@PSSH2k. Vertical black lines are expected peak positions for a 2D-hexagonal lattice. **c** Fit results for the lattice constant $a$ (assuming a 2D-hexagonal lattice) averaged over all measurements (>700 for each sample) on an absolute $a$-scale. **d** Exemplary fit results (same color code as in **c**) on a normalized $P(a)$ and $a$-scale ($\Delta a = a - \langle a \rangle$) compared to results for an AuNP@Citrate-based superlattice (gray dashed line, $d$ ~ 30 nm, functionalized with PSSH5k).

distance, number of layers, AuNP shape, diameter, and lattice type[35,36,53,54]. In consequence, an accurate control of all these parameters enables the design of tailored optical materials. Figure 4a shows the optical reflectance of supercrystals with different nanoparticle diameters. The polariton energies red shift with increasing nanoparticle diameter because of an increase in light–matter coupling strength[35]. Furthermore, the reflection dips become more pronounced because of a smaller wavelength mismatch of the light inside and outside the supercrystal. The strong dependence of the polariton energy on the nanoparticle diameter and other structural parameters implies that already slight variations within the crystal will lead to a spectral broadening or even disappearance of the polaritonic modes[36]. Figure 4b shows a comparison of two superlattices of similar geometry ($d$ ~ 30 nm, gaps ~3–4 nm): one prepared starting with AuNP@Citrate and the other one with AuNP@CTAC. Both were functionalized with PSSH5k, i.e. the only difference is the quality of the core AuNP.

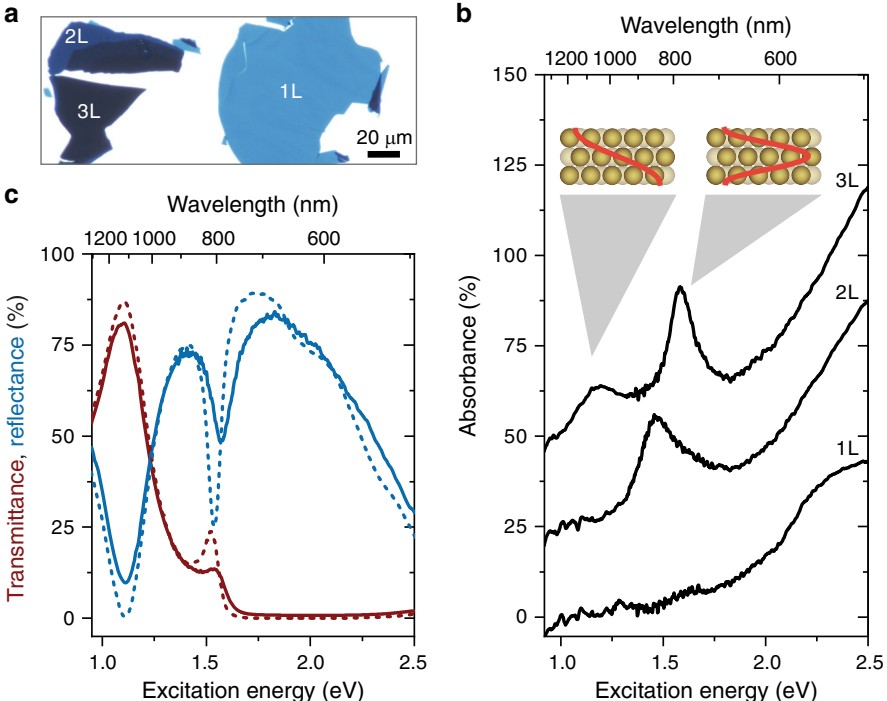

**Fig. 3 Optical properties of AuNP superlattices. a** Optical transmittance microscope image of a AuNP50@PSSH5k superlattice deposited on a glass substrate. The number of nanoparticle layers is clearly visible from the optical contrast and given by the labels. **b** Measured absorbance spectra of a AuNP50@PSSH5k monolayer (1L), bilayer (2L), and trilayer (3L). For 2L and 3L, pronounced absorbance maxima arise from the direct optical excitation of plasmon-polaritons that form standing waves in the supercrystal (see inset for 3L). With each additional layer an additional polaritonic mode is activated. The spectra are offset by 25% for clarity. **c** Measured reflectance (blue) and transmittance (red) spectra for the same AuNP trilayer as in **a** (solid lines). Simulated spectra are shown for comparison (dashed lines).

The different uniformity and crystallinity of the two lattices is clearly visible in the spectra (Fig. 4b) and in microscope images (Fig. 4c, d). The polaritonic modes broaden or even vanish for a decreasing particle and crystalline quality.

The large transmittance of light and small interparticle gaps lead to a strong near-field intensity enhancement at the gaps between the NPs throughout the entire supercrystal[4,54,63]. In combination with the tunability of the polaritonic excitations, this makes plasmonic supercrystals a promising platform for applications in sensing and spectroscopy[4,64,65]. For example, the dominant polariton mode of the trilayer in Fig. 3c matches the wavelength 785 nm of a typical laser used for Raman spectroscopy. The comparison of the Raman scattering of the bulk ligand (PSSH5k) and surface-enhanced Raman-scattering (SERS) of the ligand within the superlattice hot spots yields an enhancement factor of $\sim 3 \times 10^4$ per molecule (Supplementary Fig. 15 and Supplementary Note 1). This stresses the potential of defined AuNP superlattices for SERS and related techniques like surface-enhanced infrared-absorption (SEIRA) spectroscopy in addition to the design of polaritonic materials.

## Discussion
Large AuNP-supercrystals with hexagonal order ranging from 2D to 3D, i.e. monolayer to multilayer, were synthesized with straightforward protocols. By varying particle diameter and concentration, as well as ligand length and evaporation time, large AuNP-superlattice sheets were synthesized with varying number of layers and interparticle distances in the range of 1–8 nm. Since PSSH-thiol ligands are available with almost arbitrary molecular weights, larger gaps should be possible as well, but for strong plasmonic coupling small gaps are preferable. The obtained large scales in combination with the high degree of periodic order allow

the formation of periodically coupled plasmon-polariton modes, pointing towards applications in optical metamaterials and surface-enhanced spectroscopies. More complex superlattices (e.g. binary and including NP of different materials) can be envisaged depending on the desired application. A key point for utilizing the protocol for different NP materials and shapes will be the particle quality and a controlled surface chemistry. A complete ligand exchange and thorough purification should provide stable dispersions of the according nanomaterial in suitable solvents without additional stabilizers and at sufficiently high concentrations. The correlations of geometry and electronic and/or mechanical properties would be interesting to address in future studies. Wang et al. recently presented organic nano-floating-gate memory devices based on self-assembled AuNP-films, which greatly benefit from defined ordered structures compared to amorphous films[41]. This can reasonably be expected to be the case for many other microelectronic devices. Thus, the potential for fundamental studies with well-defined superlattice structures is quite broad. Our protocol expands the available range of precise geometries for such studies. It demonstrates that self-assembled structures obtained by wet-chemistry can compete with lithographic precision and control, and probably also provide sufficiently large areas for various devices and applications in the near future.

## Methods

**Materials**. Tetrachloroauric(III) acid (≥99.9% trace metals basis), hexadecyltrimethylammonium bromide (CTAB, ≥98%) and chloride (CTAC, ≥98%), L-ascorbic acid (reagent grade) and sodium borohydride (≥98%) were from Sigma-Aldrich (USA). Toluene (≥99.5%), tetrahydrofuran (≥99.5%), and ethanol (denat., >96%) were from VWR (USA). DEG (reagent grade) was from Merck (Germany). Thiolated polystyrenes (PSSH, PSSH2k: $M_n$ = 2000 g mol⁻¹, $M_w$ = 2300 g mol⁻¹; PSSH5k: $M_n$ = 5300 g mol⁻¹, $M_w$ = 5800 g mol⁻¹; PSSH10k: $M_n$ = 11,500 g mol⁻¹,

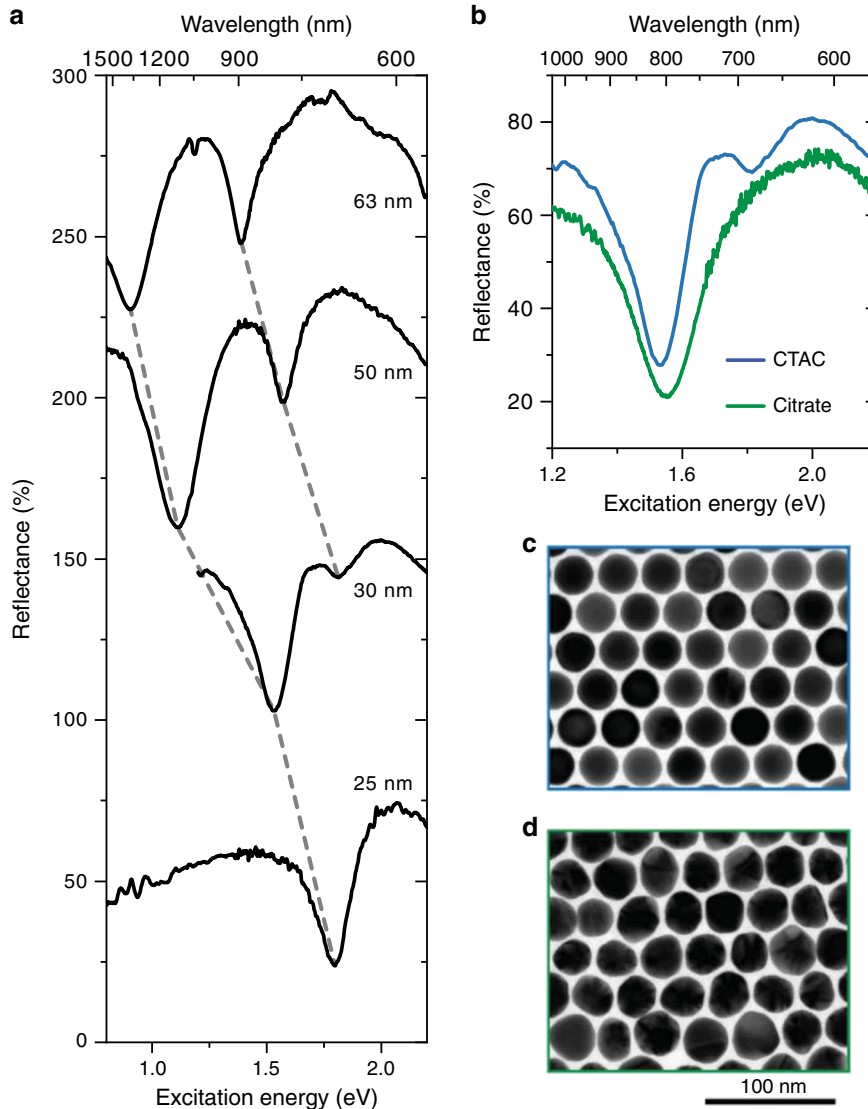

**Fig. 4 Role of structural parameters and order for the properties of plasmon-polaritons. a** Reflectance spectra of AuNP trilayers with different NP diameters (i.e. AuNP25@PSSH5k, AuNP30@PSSH5k, AuNP50@PSSH5k, AuNP60@PSSH12k—see labels). Reflectance dips that belong to the same polaritonic mode are connected by dashed lines. The spectra are offset by 75% for clarity. **b** Comparison of the reflectance spectra of 30 nm AuNP trilayer supercrystals synthesized from AuNP@CTAC (blue line) and AuNP@Citrate (green line), i.e. different core particles were coated with PSSH5k. The main reflection dip in the spectrum of the AuNP@Citrate-based superlattice is significantly broadened and the reflection dip at higher energy is not discernable. Both spectra were recorded on trilayers with the same experimental settings and were reproducible for trilayers throughout the sample. **c** TEM image of an AuNP@CTAC-based monolayer ($d = (29.7 \pm 0.9)$ nm, gaps ~3 nm). **d** TEM image of an AuNP@Citrate-based monolayer ($d = (31.8 \pm 2.4)$ nm, gaps 3–4 nm).

**Table 2 Parameters of AuNP@CTAC seeds used for AuNP@CTAC syntheses.**

| Sample | $V_{seeds}$ (µl) | $d_{seeds}$ (nm) | $c_{seeds}$ (nM) |
|---|---|---|---|
| AuNP25@CTAC | 750 | 9.5 ± 0.5 | 37 |
| AuNP35@CTAC | 100 | 9.1 ± 0.4 | 43 |
| AuNP50@CTAC | 75 | 11.1 ± 0.7 | 21 |
| AuNP60@CTAC | 45 | 11.1 ± 0.7 | 21 |

$M_{w} = 12,400$ g mol$^{-1}$) were from Polymer Source (Canada). All reagents were used without further treatment.

**AuNP@CTAC synthesis**. Gold nanoparticles (AuNP) were synthesized based on the protocol presented by Zheng et al. [55] that was upscaled to achieve sufficiently high AuNP concentrations. All solutions were in ultrapure water form (18.2 Ω). For the initial CTAB-stabilized seeds, sodium borohydride (NaBH$_4$, 600 µl, 10 mM) was quickly injected into a mixture of CTAB (9.9 ml, 100 mM) and tetrachloroauric (III) acid (HAuCl$_4$, 100 µl, 25 mM) under rapid stirring (1000 rpm) and then stirred at 300 rpm for 3 min. Afterwards, the reaction mixture was left for 3 h without agitation.

*First growth step*: 50 µl of these CTAB-seeds were mixed with ascorbic acid (1.5 ml, 100 mM) and CTAC (2.0 ml, 200 mM) at 600 rpm, followed by a one-shot-injection of HAuCl$_4$ (2.0 ml, 0.5 mM). After 15 min stirring (300 rpm), the AuNP were purified by centrifugation (20,000 × $g$ × 20 min) and the pellet redispersed in CTAC (1.0 ml, 20 mM). For an upscaled version of this first growth step, all volumes were scaled up by a factor of 10 without any other changes to the protocol. These AuNP@CTAC were used as seeds to synthesize all AuNP@CTAC with larger diameters. To this end the AuNP@CTAC were purified by two additional centrifugation steps, without changing the AuNP and CTAC concentrations, i.e. the pellets were redispersed in 20 mM CTAC.

*Second growth step*: CTAC (20 ml, 100 mM) and the desired volume of AuNP@CTAC seeds (in 20 mM CTAC) were mixed and treated with ultrasound for 10 min. The mixture was then stirred (400 rpm) in a water bath at 30 °C, ascorbic acid (1.3 ml, 10 mM) was added and after one minute the addition of

HAuCl$_4$ (20 ml, 0.5 mM) with a syringe pump at 20 ml h$^{-1}$ was started. After the addition completed, the mixture was stirred (400 rpm) for another 10 min at 30 °C and then centrifuged, and the AuNP@CTAC pellet redispersed in the same volume of water. The parameters of the AuNP@CTAC seeds (added volume $V_{seeds}$, diameter $d_{seeds}$, and seed concentration $c_{seeds}$) used for the different samples are summarized in Table 2.

**Ligand exchange.** The direct ligand exchange protocol which we used previously for AuNP@Citrate[56] could not be applied in the case of AuNP@CTAC and a modified approach had to be used. In a typical procedure 10 ml of the AuNP@CTAC solution were centrifuged twice in order to remove excess CTAC. The volume of the solution was reduced to <100 µl after the second centrifugation round and such concentrated particles were then added dropwise into a vigorously stirred (1000 rpm) solution of the respective PSSH-ligand in THF (2 ml, 0.5 mM). The resultant mixture was left to be stirred at 300 rpm for at least 2 additional hours (usually overnight) to allow for a complete replacement of CTAC by PSSH. The THF solution of AuNP@PSSH was then dried in a rotary evaporator and the particles were redispersed in toluene. Finally, two centrifugation rounds were used to remove excess PSSH ligands from the solution and to increase the particle concentration for film preparation. Residual CTAC can adversely affect ordered film formation and should be removed by repeated washing of the toluene phase with water if necessary. To this end, equal volumes of water and ethanol are added to yield a 1:1:1 mixture with the AuNP@PSSH in toluene. After thorough mixing and phase separation, the toluene phase containing the AuNP@PSSH is removed and the procedure repeated. If the phases do not separate properly, sodium chloride can be added. The final toluene solution is centrifuged twice and adjusted with toluene to the desired final concentration. UV/Vis spectroscopy measurements were used to evaluate the concentration losses in nanoparticle solutions due to the ligand exchange procedure. AuNP with diameters >25 nm were not reliably stabilized (i.e. irreversible aggregation during centrifugation steps was not prevented by the polymer shell) with PSSH2k, although one batch of stable AuNP60@PSSH2k was obtained with comparably low yield. With PSSH12k, yields (comparing the amount of initial AuNP@CTAC with the final purified AuNP@PSSH) from 65% to >95% were obtained, with PSSH5k: 40–60%. For AuNP with a diameter of 25 nm yields >90% were obtained with all three ligands.

**Self-assembly of AuNP@PSSH at the liquid–liquid interface.** The method introduced by Dong et al. was used for self-assembly of AuNP@PSSH[57]. In all experiments 100 µl of the AuNP@PSSH in toluene (prepared according the above described ligand exchange protocol) were pipetted onto 300 µl of DEG in a HDPE well (inner diameter ~1.1 cm, max. vol = 1.0 ml). The well was covered with a glass slide in order to reduce the evaporation rate of toluene. The waiting times were at least 12 h, after which toluene was evaporated and a golden film had formed on the surface of the DEG phase. The particle concentrations for film formation of different AuNP@PSSH solutions are presented in Table 1. Because of the fragility of the films and associated challenges with the film transfer, the area fraction of the different crystals (monolayer, bilayer, trilayer, and multilayer) has to be interpreted with care. As a tendency we observed that with increasing ligand length thicker crystals, i.e. more layers, are obtained, but evaporation time also plays a role. Fast evaporation (without a lid) leads preferably to monolayers with more defects, slow evaporation favors thicker crystals. A smaller number of defects upon slow evaporation was also described by Gu et al. for self-assembled monolayers of small AuNP@PSSH ($d = 5$ nm)[11]. A longer evaporation time can be achieved by covering the well with a glass slip as described and additionally by increasing the volume of AuNP@PSSH solution. The AuNP concentration did not sensitively affect the outcome of the self-assembly experiments. As a tendency, higher particle numbers lead to thicker crystals. The particles number can be increased via the concentration or the volume of drying solution. The optimum concentration to reliably obtain self-assembled AuNP films was estimated with the theoretical concentration needed for a full monolayer coverage of the available area. Even with much higher concentrations well-ordered films including monolayer and bilayer were obtained. On the other hand, too low concentrations prevent the formation of extended well-ordered films. The concentrations used herein range from 0.3 nM for AuNP60 up to 30 nM for AuNP25. Considering the 2D hexagonal structure of the films, the area of one AuNP@PSSH in the films is $A_{AuNP} = (d_{AuNP} + gap)^2 \sin(\pi/3)$, so for instance to cover 1.0 cm$^2$ with a monolayer of 35 nm AuNP with 5 nm gaps, $7.2 \times 10^{10}$ AuNP would be required, corresponding to 100 µl of a 1.2 nM solution. Accordingly, guide values for suitable AuNP concentrations can be calculated.

**Transmission electron microscopy.** TEM measurements were performed using a Jeol JEM-1011 instrument operating at 100 kV. Samples of self-assembled AuNP@PSSH films were carefully skimmed off with a carbon-coated copper grid held by a tweezer. The grid was then dried in vacuum for at least 1 h. Quantitative analyses of AuNP@CTAC size distributions and nearest-neighbor distances of AuNP@PSSH superlattices based on TEM measurements were performed with ImageJ (version 1.50i). The indicated mean values are obtained from analyses of at least three images from different spots on the samples.

**UV/vis spectroscopy.** Absorbance measurements were carried out using a Varian Cary 50 spectrometer. Quartz cuvettes (Hellma QS, Hellma, Germany) were used. The AuNP concentrations were estimated with the absorbance at 400 and 450 nm as described by Haiss et al. and Scarabelli et al.[66,67].

**Small-angle X-ray scattering.** The SAXS experiments were performed at beamline P10 at PETRA III (DESY, Hamburg). An X-ray photon energy of $E = 8$ keV was used. The X-ray beam was focused by compound refractive lenses[68] to 2.8 µm × 1.7 µm ($h \times v$). The detector (Dectris Eiger X 4M) was placed 5 m downstream of the sample, covering a $q$-range from 0.014 to 1.15 nm$^{-1}$. The samples were mounted in a vacuum chamber connected to an evacuated flight path. SAXS patterns were taken from different regions of the samples in scanning mode. Therein, scanning steps of 5–10 µm step size were typically chosen in horizontal and vertical directions normal to the X-ray path. At least 2000 individual patterns were measured from each sample in this way, obtaining >700 hits that were used for further analysis (cf. Fig. 2a: some measurements just cover TEM-grid without sample). From the patterns, the lattice constants $a$ were obtained from the position of the indexed Bragg reflections. The distributions $P(a)$ were calculated from the histograms of $a$. The interparticles distances (gaps) based on SAXS (cf. Table 1) were determined by substracting the AuNP diameters from particle form factor fits from the lattice parameters $a$.

**Optical microscopy.** The optical spectra of the gold nanoparticle trilayer were recorded with a home-build micro-absorbance spectrometer. Details of the setup can be found in ref. [36]. A supercontinuum laser (Fianium, SC-400-4) was used as a white light source. The light was guided through a linear polarizer into an inverted optical microscope (Olympus, IX71) and focused with an ×100 objective (Leica HCX PL Fluotar ×100) onto the sample. We used objectives with a numerical aperture of NA = 0.9 to focus light to a diffraction-limited spot on the sample. This corresponds to a maximum angle of incidence of 64° outside the supercrystal. The maximum angle of incidence inside the supercrystal is with 5–13°, much smaller and close to normal incidence because of the large effective index of refraction, $n_{eff} = 4$–10, of the supercrystals[35]. The laser power was kept below 200 µW. The AuNP films were transferred to a glass substrate for optical characterization. The layer number was identified through the optical contrast in transmission (cf. Supplementary Fig. 14). The sample was moved in $x$-direction and $y$-direction with a combination of a motorized stage and a piezo stage. The transmitted light was collected with a second ×100 objective (Olympus MPlan FL N BD ×100) and coupled through a fiber (Ocean Optics, QP600-2-UV-BX for 450–900 nm and BIF600-VIS–NIR for 900–1600 nm) into a spectrometer (Avantes, AvaSpec 3648 for 450–900 nm and AvaSpec NIR512 for 900–1600 nm). In a subsequent measurement, the back-reflected light was separated from the incoming light beam with a beam splitter (ThorLabs, BSW26R for 450–900 nm and BSW29R for 900–1600 nm) and detected with the same combination of fibers and spectrometers as for the transmitted light. The spectra of the VIS and NIR ranges were pinned together.

**Finite-difference in the time-domain (FDTD) simulations.** For the FDTD simulations in Fig. 3 the commercial software package Lumerical FDTD Solutions was used as described recently[36,54]. Spherical NPs with 50 nm diameter were packed into a hexagonal lattice with 1.6 nm interparticle gaps. Three layers were stacked in the $abc$ sequence of an fcc crystal to construct a trilayer. A mesh override region with 0.5 nm cells was used to resolve the geometry of the supercrystal. The dielectric function of gold was modeled by fitting experimental data from Olmon et al. [69] and the polystyrene ligand molecules were modeled as a dielectric medium with refractive index $n = 1.4$ filling the space between the NPs. The FDTD simulation region was chosen as the unit cell of the supercrystal with periodic boundary conditions along $x$ and $y$. Perfectly matched layers were used along $z$. Light was injected along $z$ with a broadband plane wave source. The transmittance and reflectance were recorded with power monitors behind the nanoparticle layers and behind the source.

## Data availability

The data that support the findings of this study are available from the corresponding author upon reasonable request.

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

## Acknowledgements

This work is supported by the Cluster of Excellence'Advanced Imaging of Matter' of the Deutsche Forschungsgemeinschaft (DFG)-EXC 2056-project ID 390715994. F.S. was supported by the DFG via the project SCHU 3019/2-1. Y.O., N.S.M., and S.R. were supported by the European Research Council (ERC) under grant DarkSERS (772108). O.P. was supported by the Czech Science Foundation (Grant number: 18-07977Y GACR) and the Herman & Else Schnabel Stiftung. We acknowledge DESY (Hamburg, Germany), a member of the Helmholtz Association HGF, for the provision of experimental facilities. Parts of this research were carried out at beamline P10 of PETRA III. The authors thank Dr. Andreas Meyer for help with the nearest-neighbor-analyses and Finn Dobschall for supporting work on AuNP syntheses, as well as Sabrina Jürgensen and Robert Schön for help with the SEM images.

## Author contributions

F.S. developed the synthesis protocol. F.S. and O.P. synthesized AuNP and AuNP superlattices, F.S. did the TEM measurements, F.S. and O.P. analyzed the TEM data, F.S., F.L., and F.W. performed the SAXS measurements, F.L. and F.W. analyzed the SAXS data, N.S.M. and Y.O. did the optical measurements and calculations including SERS. F.S., H.L., and S.R. supervised the project. All authors discussed the results and implications and commented on the manuscript at all stages. F.S. and H.L. wrote the manuscript with contributions of all authors. All authors have given approval to the final version of the manuscript.

## Competing interests

The authors declare no competing interests.
