## [Peer Review File · Nature Communications]

REVIEWER COMMENTS

Reviewer #1 (Remarks to the Author):

The paper by Schulz et al. report on the fabrication of plasmonic superlattices and their optical characterization. The results shown in the manuscript are really interesting since they have combined the synthesis of gold spheres with high monodispersity together with slow evaporation of toluene over a DEG leading to plasmonic superlattices that in agreement with the simulations support collective plasmon polariton modes. The manuscript is interesting and the results shown clear but there are a number of points that need to be clarified prior to support its acceptance.

- To what extent can the number of layers be controlled?, which are the critical parameters for that?.
- When reporting the optical properties of the layers I think that there a lot of missing information; in Fig 4 is reported the reflectance of trilayer, the same data should be given a single monolayer also for a bilayer (together with the FDTD simulation). Which is the influence of the PSS length on the optical response of the plasmon polariton?.

It would be interesting also provide X axis in terms of nm apart from eV. Reporting also the absorbance or the transmittance will help the reader.

Related with the optical properties, and considering the tight control that the authors reported I think that a similar data to that mentioned above should be also give for the different sizes and the different PSS sizes.

In summary, I think that the manuscript could be considered for publication but a more deep study and analysis of the optical properties of the different systems that the authors claimed can be fabricated need to be reported.

Reviewer #2 (Remarks to the Author):

The manuscript by Schulz et al. titled "Structural order in plasmonic superlattices" addresses two of the fundamental issues in the field of nanoparticle (NP) self-assembly, namely (i) the fabrication of organized, possibly defect-free two- and three-dimensional arrays of NPs with long-range order and (ii) the subsequent control of the interparticle separation distances, so as to modulate the collective properties of the ensemble, such as, e.g., its plasmonic/polaritonic ones. Common strategies to achieve these goals involve functionalization of the NP surfaces (i) with polymers of different lengths (as in this study), (ii) with DNA [see, e.g., Liu et al., *Science* 351, 582 (2016)], and (iii) with dendritic molecules of various generations [see, e.g., Malassis et al., *Nanoscale* 8, 13192 (2016)].

As acknowledged by the authors in their manuscript, many related studies have been reported so far, and the employed experimental techniques they use are a priori routine, which may suggest that the present work lacks originality and novelty. However, the presented results constitute in my opinion a very interesting contribution to the field, as Schulz et al. obtain, to the best of my knowledge, an unprecedented long-range order of the gold NP lattice. These results pave the way to promising prospects for, e.g., plasmonic metamaterials. Additionally, the manuscript is well written and reads well. Consequently, I would in principle recommend publication in *Nature Communications*. However, there are some elements of the discussion that I believe are not fully convincing and which should be clarified before the final publication of the manuscript.

1/ In particular, the different interparticle gaps measured for NPs of various sizes but bearing the same ligand are ascribed to stronger attractive interactions between Au NPs as their diameter is increased (see the discussion around Fig. 2). As recognized by the authors (see lines 133 and 134), this may be true for series with PSSH12k, where a linear gap-diameter relation is found (see Fig. 2),

but not for PSSH5k: as the NP diameter increases from ca. 25 to 60 nm, the gap varies in a sawtooth manner for the series PSSH5k (3.3, 5.1, 1.3, and 2.2 nm, see Table 1). The results for PSSH2k are not conclusive, since only two samples were considered. I therefore wonder if Fig. 2, which only presents results for PSSH12k, is not slightly misleading. While it is plausible that van der Waals forces are at work in explaining the data in Fig. 2, why should it also be the mechanism able to interpret the PSSH5k series, given the nonmonotonic gap-size relation in that case?

2/ For the larger NPs in Table 1, the gaps lie in the range of the mean standard deviation found for the NP diameter. I therefore wonder if these results obtained by TEM measurements are meaningful for such large NPs. Probably, these results should be compared with the lattice constant a obtained from the SAXS ones. It would also be useful in Table 1 to provide the results of the SAXS measurements (when available, since apparently only 5 samples out of 10 were measured).

In addition to my two comments above, I also have several remarks and questions on the present manuscript that should certainly be addressed by the authors:

3/ Refs. 9 and 10 are exactly the same.

4/ On line 46, the authors state that "According to simulations, the coupling strength (...)". To which simulations are they referring to? Those in Refs. 25-27? This is slightly ambiguous.

5/ On line 93, the authors mention "Murray's group" when introducing the work presented in Ref. 44. I must admit that I find this wording quite unfortunate and slightly discourteous for the 1st author of Ref. 44 (who probably did most of the work). I would rather for instance employ the wording "Dong et al." which gives full credit to all contributing authors, and not only to the group leader.

6/ Fig. 1b is, at least on a printed version of the manuscript, way too small, so that it is very difficult to see the actual sample.

7/ About the measured reflectance spectrum shown in Fig. 4, the angle of incidence should probably be specified.

8/ On line 185, the authors claim that "[their] experimental spectra are in excellent agreement" with FDTD simulations obtained with the commercial software package Lumerical. However, even though the simulation (dashed black line in Fig. 4) is in fairly good qualitative agreement with the measurement (solid blue line), I would not say that the agreement is "excellent", as there are still significant quantitative differences between the two curves (see, in particular, the "down" peak feature at around 1.5 eV, where the measured reflectance is of about 25%, while the simulation predicts 50%, that is, a factor of 2 larger).

9/ Still about the results presented in Fig. 4: the simulations predict a vanishing reflectance for an excitation energy around 1.1 eV, while the measured reflectance is finite and equals roughly 10%. Could the authors comment on this qualitative disagreement?

10/ In Fig. 5, one can observe a small energy shift between the two curves for the polaritonic mode at around 1.5 eV. Can it be due to the two different dielectric environments produced by the two different ligands, and the subsequent effect on the localized plasmon resonance frequency of the individual NPs (which itself influence the collective polaritonic modes)?

Reviewer #3 (Remarks to the Author):

“Structural Order in Plasmonic Superlattices” by F. Schultz et al

The authors, propose and deeply investigate highly ordered plasmonic arrays and multilayers of these arrays. A wide variety of arrays of plasmonic NPs have been proposed, synthesized, and studied by a large number of groups. From small to large NPs, from small to large lattice parameters, While some groups succeeded in obtaining well controlled NP separation, others succeeded in the fabrication of large scale arrays or highly uniform NP shape/size. Here, the authors succeeded at fabrication highly ordered, almost defect free of highly NPs with very homogeneous size and shape, and this on a large scale.

In itself this warrants publication of these results as the proposed fabrication procedure will have a strong impact on a large community seeking robust NP-based platform for a wide range of application.

I do think the authors could have done a slightly better job at placing their work in context as major pioneer works are not cited and/or discussed (e.g.: work by Mirkin on DNA-based superlattices, and Odom on 2D NP arrays), including work on bi-metallic superlattices (Ross et al, Nat. Comm. 2016). The recent review paper by Liz Marzan (Acc. Chem. Res 2019) is also to be cited along with the 2006 Nature paper by Chris Murray.

Also, although the authors compare their experimental reflectance measurement to some FDTD simulation, there's very little comparison to theoretical predictions of the superlattice properties. If the lattice are highly ordered, with very little size, shape, and gap variation throughout the array, a more robust and systematic comparison with theory/simulation would strengthen the authors' conclusions even more. I know the authors have already recently published theory papers on these systems, but a better discussion and closer comparison between the two would be very interesting here.

Although the authors reference the FDTD method section to previous papers, I think it is important that the authors briefly summarize the numerical protocol and model in this section so that the reader can quickly get a sense of how the numerical modeling is done.

Apart from these, the results are very convincing and show tremendous potential for future applications. I personally wondering how straight forward it will be to extend this approach to other materials (Ag, Al, Pd,...). Also, although the authors briefly mention it at the end, I would recommend the author to briefly discuss how their approach could be expanded to anisotropic NPs such as cubes, rod/ellipsoids, prisms for instance.

In short, I strongly recommend publication of this work in Nature Communication.

Point-by-point response to the reviewers' comments

We thank all reviewers for their thorough evaluation of the manuscript and the supportive feedback. We have no objections and considered all points raised by the reviewers in our revision as detailed below. In particular we include more optical characterization data as recommended by all reviewers.

Reviewer #1 (Remarks to the Author):

The paper by Schulz et al. report on the fabrication of plasmonic superlattices and their optical characterization. The results shown in the manuscript are really interesting since they have combined the synthesis of gold spheres with high monodispersity together with slow evaporation of toluene over a DEG leading to plasmonic superlattices that in agreement with the simulations support collective plasmon polariton modes. The manuscript is interesting and the results shown clear but there are a number of points that need to be clarified prior to support its acceptance.

- To what extent can the number of layers be controlled?, which are the critical parameters for that?.

Answer: An exact control of the number of layers, that is definitely highly desirable, has not been achieved. However, we already have indications of the relevant parameters. In the revised version, we point that out more clearly and include a reference to the more detailed discussion in the method section by the following statement:

"The number of layers cannot be controlled perfectly, but as a tendency slow evaporation and an increasing number of particles in the drying solution seem to favor higher fractions of multilayer superlattices. Accordingly, a fast evaporation and low AuNP concentrations favor the formation of monolayers. Quantitative statements are not possible because the exact fractions of the different layer numbers within one sample are not accessible."

- When reporting the optical properties of the layers I think that there is a lot of missing information; in Fig 4 is reported the reflectance of trilayer, the same data should be given for a single monolayer also for a bilayer (together with the FDTD simulation). Which is the influence of the PSS length on the optical response of the plasmon polariton?.

Answer: In the revised version we provide more optical characterization data in Figures 3 and 4 as requested, including spectra for different numbers of layers and different particle diameters. We also revised the according discussion in the main text and include new references to guide readers to more exhaustive descriptions of the optical properties, which are out of scope of this manuscript. These include discussions of the role of the gap size for the optical response.

It would be interesting also provide X axis in terms of nm apart from eV. Reporting also the absorbance or the transmittance will help the reader.

Answer: In the revised figures 3 and 4 nm-axes, as well as absorbance and transmittance spectra are included.

Related with the optical properties, and considering the tight control that the authors reported I think that a similar data to that mentioned above should be also give for the different sizes and the different PSS sizes.

In summary, I think that the manuscript could be considered for publication but a more deep study and analysis of the optical properties of the different systems that the authors claimed can be fabricated need to be reported.

Answer: In the revised version, we include additional optical characterization and a more thorough discussion of the optical properties as requested. References to recent works, that are focussed primarily on the optical properties of AuNP superlattices and -crystals are also provided.

Reviewer #2 (Remarks to the Author):

The manuscript by Schulz et al. titled "Structural order in plasmonic superlattices" addresses two of the fundamental issues in the field of nanoparticle (NP) self-assembly, namely (i) the fabrication of organized, possibly defect-free two- and three-dimensional arrays of NPs with long-range order and (ii) the subsequent control of the interparticle separation distances, so as to modulate the collective properties of the ensemble, such as, e.g., its plasmonic/polaritonic ones. Common strategies to achieve these goals involve functionalization of the NP surfaces (i) with polymers of different lengths (as in this study), (ii) with DNA [see, e.g., Liu et al., Science 351, 582 (2016)], and (iii) with dendritic molecules of various generations [see, e.g., Malassis et al., Nanoscale 8, 13192 (2016)].

Answer: Reviewer 3 suggests to include some additional references to pioneering work. Because it is not possible to adequately cover all existing studies in this large field, we tried to provide the reader with references to topical reviews. However, we also added some references to acknowledge the pioneering work with DNA-based and lithographic structures as suggested by Reviewer 3 and included the references pointed out here. The statement in the introduction now reads:

Common strategies to produce such structures include lithography-based nanofabrication,¹⁴⁻¹⁶ and self-assembly of NPs functionalized with polymers,¹⁰ DNA¹⁷⁻²² or dendritic molecules.^{23,24}

As acknowledged by the authors in their manuscript, many related studies have been reported so far, and the employed experimental techniques they use are a priori routine, which may suggest that the present work lacks originality and novelty. However, the presented results constitute in my opinion a very interesting contribution to the field, as Schulz et al. obtain, to the best of my knowledge, an unprecedented long-range order of the gold NP lattice. These results pave the way to promising prospects for, e.g., plasmonic metamaterials. Additionally, the manuscript is well written and reads well. Consequently, I would in principle recommend publication in Nature Communications. However, there are some elements of the discussion that I believe are not fully convincing and which should be clarified before the final publication of the manuscript.

1/ In particular, the different interparticle gaps measured for NPs of various sizes but bearing the same ligand are ascribed to stronger attractive interactions between Au NPs as their diameter is

increased (see the discussion around Fig. 2). As recognized by the authors (see lines 133 and 134), this may be true for series with PSSH12k, where a linear gap-diameter relation is found (see Fig. 2), but not for PSSH5k: as the NP diameter increases from ca. 25 to 60 nm, the gap varies in a sawtooth manner for the series PSSH5k (3.3, 5.1, 1.3, and 2.2 nm, see Table 1). The results for PSSH2k are not conclusive, since only two samples were considered. I therefore wonder if Fig. 2, which only presents results for PSSH12k, is not slightly misleading. While it is plausible that van der Waals forces are at work in explaining the data in Fig. 2, why should it also be the mechanism able to interpret the PSSH5k series, given the nonmonotonic gap-size relation in that case?

Answer: This is a valid point. Since we can just speculate at this point about the additional contributing parameters, we changed the statement in the main text and moved the figure to the Supplementary Material (now Figure S13). We also changed the figure to include the data for all ligand lengths. We think that this way we can report the doubtless correlation of AuNP diameter and gap size for the ligand PSSH12k while being transparent about the less clear observations for the smaller ligands. The statement in the main text now reads:

“For PSSH12k a correlation of gap distance with particle diameter was observed that points to the increasing attractive van-der-Waals-potentials between the AuNP-cores (Supplementary Fig. 13).⁶² However, with PSSH5k the trend was not that clear, indicating that additional parameters of the crystallization affect the gaps. For PSSH2k just two samples can be compared due to the limited stabilization provided by this small ligand.”

2/ For the larger NPs in Table 1, the gaps lie in the range of the mean standard deviation found for the NP diameter. I therefore wonder if these results obtained by TEM measurements are meaningful for such large NPs. Probably, these results should be compared with the lattice constant obtained from the SAXS ones. It would also be useful in Table 1 to provide the results of the SAXS measurements (when available, since apparently only 5 samples out of 10 were measured).

Answer: Indeed, the accuracy of the TEM analysis for the larger particles is lower, because they exhibit a lower uniformity and larger absolute standard deviation of the mean diameter. The gaps can be clearly observed in the TEM measurements, but meaningful values for the gap sizes are harder to obtain, justifying the complementary SAXS characterization. We think the referee raises an important point here and accordingly included SAXS data to Table 1 and the following statement in the main text:

“The gaps obtained by SAXS deviate up to a few nm from those obtained by TEM analysis. This observation points at the limits of TEM analysis accuracy, especially for the larger AuNP which are slightly less uniform (cf. Supplementary Figures 2-11).⁵⁸ Consequently, complementary SAXS measurements for comparison are recommendable for the characterization of superlattices in general.”

The new reference 58 discusses the accuracy of TEM characterization of nanomaterials in detail.

In addition to my two comments above, I also have several remarks and questions on the present manuscript that should certainly be addressed by the authors:

3/ Refs. 9 and 10 are exactly the same.

Answer: The duplicate was removed.

4/ On line 46, the authors state that "According to simulations, the coupling strength (...)". To which simulations are they referring to? Those in Refs. 25-27? This is slightly ambiguous.

Answer: We changed the wording and include the citations directly with the statements now to avoid this ambiguity. A new reference (35) to recent experimental work is now included.

5/ On line 93, the authors mention "Murray's group" when introducing the work presented in Ref. 44. I must admit that I find this wording quite unfortunate and slightly discourteous for the 1st author of Ref. 44 (who probably did most of the work). I would rather for instance employ the wording "Dong et al." which gives full credit to all contributing authors, and not only to the group leader.

Answer: We changed the wording accordingly.

6/ Fig. 1b is, at least on a printed version of the manuscript, way too small, so that it is very difficult to see the actual sample.

Answer: We changed the figure accordingly.

7/ About the measured reflectance spectrum shown in Fig. 4, the angle of incidence should probably be specified.

Answer: We added the following statement to the method section:

We used objectives with a numerical aperture of $NA = 0.9$ to focus light to a diffraction limited spot on the sample. This corresponds to a maximum angle of incidence of 64° outside the supercrystal. The maximum angle of incidence inside the supercrystal is with $5-13^\circ$, much smaller and close to normal incidence because of the large effective index of refraction, $n_{eff} = 4-10$, of the supercrystals.³⁵

8/ On line 185, the authors claim that "[their] experimental spectra are in excellent agreement" with FDTD simulations obtained with the commercial software package Lumerical. However, even though the simulation (dashed black line in Fig. 4) is in fairly good qualitative agreement with the measurement (solid blue line), I would not say that the agreement is "excellent", as there are still significant quantitative differences between the two curves (see, in particular, the "down" peak feature at around 1.5 eV, where the measured reflectance is of about 25%, while the simulation predicts 50%, that is, a factor of 2 larger).

Answer: We changed the wording to "good" agreement.

9/ Still about the results presented in Fig. 4: the simulations predict a vanishing reflectance for an excitation energy around 1.1 eV, while the measured reflectance is finite and equals roughly 10%. Could the authors comment on this qualitative disagreement?

Answer: The simulated reflectance dips are more pronounced and spectrally narrower, because of an inhomogeneous broadening in the measured spectra. This inhomogeneous broadening occurs because of slight variations of the diameters, gaps and nanoparticle shapes on a length scale much smaller than the spot size of the laser. This is also apparent from Figure 4b, which compares the spectra of a highly ordered sample and a less ordered sample.

10/ In Fig. 5, one can observe a small energy shift between the two curves for the polaritonic mode at around 1.5 eV. Can it be due to the two different dielectric environments produced by the two different ligands, and the subsequent effect on the localized plasmon resonance frequency of the individual NPs (which itself influence the collective polaritonic modes)?

Answer: We apologize for this misunderstanding. In fact, both AuNP batches were functionalized with the same ligand PSSH5k. We use @CTAC and @Citrate to indicate the different synthesis strategies for the AuNP core particles, that result in different particle qualities. After functionalization with PSSH5k there should be only minor differences in the dielectric environments provided by the coatings, if any. A possible explanation for the energy shift is the variation of interparticle gap sizes and diameters and the different uniformities of the samples. Figure 4a in the revised manuscript now shows the effect of the nanoparticle diameter on the spectral position, illustrating the strong dependence. In the rewritten discussion and figure caption we point out more clearly, that different core AuNP, but the same coatings were used for the comparison shown in Figure 4b (Figure 5 in the previous version).

Reviewer #3 (Remarks to the Author):

“Structural Order in Plasmonic Superlattices” by F. Schultz et al

The authors, propose and deeply investigate highly ordered plasmonic arrays and multilayers of these arrays. A wide variety of arrays of plasmonic NPs have been proposed, synthesized, and studied by a large number of groups. From small to large NPs, from small to large lattice parameters, While some groups succeeded in obtaining well controlled NP separation, others succeeded in the fabrication of large scale arrays or highly uniform NP shape/size. Here, the authors succeeded at fabrication highly ordered, almost defect free of highly NPs with very homogeneous size and shape, and this on a large scale.

In itself this warrants publication of these results as the proposed fabrication procedure will have a strong impact on a large community seeking robust NP-based platform for a wide range of application.

I do think the authors could have done a slightly better job at placing their work in context as major pioneer works are not cited and/or discussed (e.g.: work by Mirkin on DNA-based superlattices, and Odom on 2D NP arrays), including work on bi-metallic superlattices (Ross et al, Nat. Comm. 2016). The recent review paper by Liz Marzan (Acc. Chem. Res 2019) is also to be cited along with the 2006 Nature paper by Chris Murray.

Answer: We included the suggested references and the references mentioned by Reviewer 2.

Also, although the authors compare their experimental reflectance measurement to some FDTD simulation, there's very little comparison to theoretical predictions of the superlattice properties. If the lattice are highly ordered, with very little size, shape, and gap variation throughout the array, a more robust and systematic comparison with theory/simulation would strengthen the authors' conclusions even more. I know the authors have already recently published theory papers on these

systems, but a better discussion and closer comparison between the two would be very interesting here.

Answer: We added optical characterization data as requested and revised the entire discussion of the optical properties. Additionally we include references to more detailed quantum optical descriptions of the AuNP superlattices' optical properties.

Although the authors reference the FDTD method section to previous papers, I think it is important that the authors briefly summarize the numerical protocol and model in this section so that the reader can quickly get a sense of how the numerical modeling is done.

Answer: The description of the FDTD simulations is now included in the methods section.

Apart from these, the results are very convincing and show tremendous potential for future applications. I personally wondering how straight forward it will be to extend this approach to other materials (Ag, Al, Pd,...). Also, although the authors briefly mention it at the end, I would recommend the author to briefly discuss how their approach could be expanded to anisotropic NPs such as cubes, rod/ellipsoids, prisms for instance.

Answer: We agree that these are exciting prospects. Every different material and shape will require thorough optimization studies on its own for sure. We think that apart from the NPs' quality a controlled surface chemistry will be crucial, because it is the key for a high purity of the nanomaterial. E.g. for the presented system we found that residual CTAC is hindering well-ordered self-assembly, as mentioned in the method section. We therefore added the following statements to the concluding discussion:

A key point for utilizing the protocol for different NP materials and shapes will be the particle quality and a controlled surface chemistry. A complete ligand exchange and thorough purification should provide stable dispersions of the according nanomaterial in suitable solvents without additional stabilizers and at sufficiently high concentrations.

In short, I strongly recommend publication of this work in Nature Communication.

REVIEWERS' COMMENTS:

Reviewer #1 (Remarks to the Author):

The revised manuscript by Schulz has taken into consideration all the concerns raised by this reviewer. I think that the manuscript has been greatly improved, especially the section devoted to the optical characterization of the superlattices. Therefore, I support its acceptance in Nature Communications. I strongly encourage the authors to try to work on the scalability of the process together with a tight control on the number of nanoparticle layers that can be achieved.

Reviewer #2 (Remarks to the Author):

In their rebuttal, Schulz et al. convincingly answered all of my points. I therefore recommend the publication of the revised version in Nature Communications.